# Targeting Yezo Virus Structural Proteins for Multi-Epitope Vaccine Design Using Immunoinformatics Approach

**DOI:** 10.3390/v16091408

**Published:** 2024-09-03

**Authors:** Sudais Rahman, Chien-Chun Chiou, Mashal M. Almutairi, Amar Ajmal, Sidra Batool, Bushra Javed, Tetsuya Tanaka, Chien-Chin Chen, Abdulaziz Alouffi, Abid Ali

**Affiliations:** 1Department of Zoology, Abdul Wali Khan University, Mardan 23200, Khyber Pakhtunkhwa, Pakistan; 2Department of Dermatology, Ditmanson Medical Foundation Chia-Yi Christian Hospital, Chiayi 600, Taiwan; 3Department of Pharmacology and Toxicology, College of Pharmacy, King Saud University, Riyadh 11451, Saudi Arabia; 4Department of Biochemistry, Abdul Wali Khan University, Mardan 23200, Khyber Pakhtunkhwa, Pakistan; 5Laboratory of Animal Microbiology, Graduate School of Agricultural Science/Faculty of Agriculture, Tohoku University, 468-1 Aramaki Aza Aoba, Aoba-ku, Sendai 980-8572, Japan; 6Department of Pathology, Ditmanson Medical Foundation Chia-Yi Christian Hospital, Chiayi 600, Taiwan; 7Department of Cosmetic Science, Chia Nan University of Pharmacy and Science, Tainan 717, Taiwan; 8Ph.D. Program in Translational Medicine, Rong Hsing Research Center for Translational Medicine, National Chung Hsing University, Taichung 402, Taiwan; 9Department of Biotechnology and Bioindustry Sciences, College of Bioscience and Biotechnology, National Cheng Kung University, Tainan 701, Taiwan; 10King Abdulaziz City for Science and Technology, Riyadh 12354, Saudi Arabia

**Keywords:** Yezo virus, immunoinformatics, multi-epitope vaccine, molecular docking, molecular dynamic simulation

## Abstract

A novel tick-borne orthonairovirus called the Yezo virus (YEZV), primarily transmitted by the *Ixodes persulcatus* tick, has been recently discovered and poses significant threats to human health. The YEZV is considered endemic in Japan and China. Clinical symptoms associated with this virus include thrombocytopenia, fatigue, headache, leukopenia, fever, depression, and neurological complications ranging from mild febrile illness to severe outcomes like meningitis and encephalitis. At present, there is no treatment or vaccine readily accessible for this pathogenic virus. Therefore, this research employed an immunoinformatics approach to pinpoint potential vaccine targets within the YEZV through an extensive examination of its structural proteins. Three structural proteins were chosen using specific criteria to pinpoint T-cell and B-cell epitopes, which were subsequently validated through interferon-gamma induction. Six overlapping epitopes for cytotoxic T-lymphocytes (CTL), helper T-lymphocytes (HTL), and linear B-lymphocytes (LBL) were selected to construct a multi-epitope vaccine, achieving a 92.29% coverage of the global population. These epitopes were then fused with the 50S ribosomal protein L7/L12 adjuvant to improve protection against international strains. The three-dimensional structure of the designed vaccine construct underwent an extensive evaluation through structural analysis. Following molecular docking studies, the YEZV vaccine construct emerged as a candidate for further investigation, showing the lowest binding energy (−78.7 kcal/mol) along with favorable physiochemical and immunological properties. Immune simulation and molecular dynamics studies demonstrated its stability and potential to induce a strong immune response within the host cells. This comprehensive analysis indicates that the designed vaccine construct could offer protection against the YEZV. It is crucial to conduct additional in vitro and in vivo experiments to verify its safety and effectiveness.

## 1. Introduction

The Yezo virus (YEZV), belonging to the genus orthonairovirus within the nairoviridae, contains circular single-stranded RNA (ssRNA) [1]. The YEZV is approximately 80–120 nm in diameter, with proteins constituting 50% of their mass and lipids making up 20–30%. The ribonucleocapsid is a long, slender structure, about 200–250 nm in length and 1.8–2.5 nm in width. The nucleocapsids are covered in a single casing with projections formed from glycoproteins extending from their surface [2]. The entire length of the genome is about 17,100–22,800 nucleotides and is divided into three parts: large, medium, and small. The large fragment that encodes the viral polymerase ranges from 11,000–14,400 (11–14.4 kb) nucleotides in length. The middle portion, which encodes for glycoproteins, is about 4400–6300 (4.4–6.3 kb) nucleotides in length [3]. The segment that encodes for the nucleocapsid proteins, measures approximately 1700 to 2200 (1.7–2.2 kb) nucleotides [4]. The YEZV poses a significant threat to human health [5]. This ssRNA virus is primarily transmitted to humans through the bite of *Ixodes persulcatus* ticks [6]. Recently, the discovery of the YEZV in Japan [7] and China [8] has drawn particular apprehension due to its potential to induce a variety of health complications, including low platelet counts, reduced white blood cells, and neurological problems [9]. The spectrum of YEZV infections extends from mild febrile illnesses to severe neurological disorders such as meningitis, acute flaccid paralysis, and encephalitis [10]. While many infections may remain asymptomatic or exhibit nonspecific symptoms, severe cases, particularly in elderly immunocompromised individuals, can lead to significant health complications and even fatality [11]. Given the increasing incidence of the YEZV and the limited availability of effective vaccines, the YEZV is targeted for an immunoinformatics-based vaccine design to combat and effectively control its measures.

Advancements in bioinformatics have led to the discovery of numerous computational techniques that deal with the identification of potential targets for vaccine and drug discovery against the pathogen [12]. Extensive computational analyses have been conducted to identify potential vaccine candidates and therapeutic targets crucial for combating a variety of pathogens [13,14,15]. Immunoinformatics has emerged as a particularly promising approach, demonstrating exceptional capability in designing precise, stable, and multi-epitope vaccine constructs [16]. Multi-epitope vaccines are characterized by their high safety profile and cost-effectiveness as they elicit both B-cell and T-cell immune responses. This capability allows them to generate robust and long-lasting immunity against specific pathogens. To optimize their effectiveness, it is crucial to predict how immunogenic epitopes interact with the major histocompatibility complex (MHC) [17]. In this research, we employed immunoinformatics and computational methodologies to design a vaccine candidate against the YEZV, incorporating the most antigenic and immunogenic epitopes derived from glycoprotein (GP), RNA-directed RNA polymerase (L), and nucleoproteins (NP).

## 2. Materials and Methods

This study utilized systematic methods to design a vaccine construct for the Yezo virus, as outlined in Figure 1.

### 2.1. Dataset Collection and Its Filtration

The proteome of the Yezo virus (YEZV) was collected from the NCBI (https://www.ncbi.nlm.nih.gov/) (Taxonomic ID: 2825847) and verified for accuracy using the Virus Pathogen Resource (ViPR) database (https://www.bv-brc.org/). To eliminate redundancy, CD-HIT with an E-value threshold of 0.8 was employed [18]. Subsequently, a BLASTp search on the NCBI was conducted to assess its homology with human proteins [19]. Further filtration of the viral proteins was evaluated to check the antigenicity utilizing the Vaxijen v2.0 “http://www.ddg-pharmfac.net/vaxijen/VaxiJen/VaxiJen.html (accessed on 8 May 2024)” with a value set at 0.4. An allergenicity screening was performed using the AllerTop “https://www.ddg-pharmfac.net/AllerTOP/ (accessed on 15 May 2024)”, and the toxicity was evaluated using Toxinpred 3.0 “https://webs.iiitd.edu.in/raghava/toxinpred/algo.php (accessed on 25 May 2024)”.

### 2.2. Prediction of Cytotoxic T-Lymphocyte (CTL) Epitopes

The MHC-I binding server at the immune epitope database was used to forecast the CTL epitopes [20]. NetMHCpan 4.1, a neural network method incorporating HLA alleles as a reference set, was utilized to forecast CTL epitopes for a wide range of populations [21]. The epitopes were evaluated according to their IC_50_ values, where a lower IC_50_ suggests a stronger MHC-I binding [22]. The threshold of IC_50_ ≤ 100 nM was selected because it represents high-affinity binding between the epitope and the MHC molecule, indicating the strong potential for immune recognition. The predicted epitopes were then assessed for antigenicity, allergenicity, and toxicity analysis [23].

### 2.3. Predicting Helper T-Lymphocyte (HTL) Epitopes and Evaluating Interferon-Gamma (IFN-γ) Induction

NetMHCIIpan 4.1 in the IEDB MHC-II binding server was utilized to forecast HTL epitopes, focusing on the human HLA-DR locus [24]. The 15-mer epitopes were sorted according to their IC_50_ values. Predicted HTL epitopes with an IC_50_ of less than 100 nM were then evaluated for their capacity to induce interferon-gamma IFN-γ), interleukin-4 (IL-4), and interleukin-10 (IL-10) secretion using the IFN-epitope server “https://webs.iiitd.edu.in/raghava/ifnepitope/help.php (accessed on 5 June 2024)”, IL4Pred “https://webs.iiitd.edu.in/raghava/il4pred/ (accessed on 10 June 2024)”, and IL10Pred “https://webs.iiitd.edu.in/raghava/il10pred/algo.php (accessed 11 June 2024)” web servers. IFN-γ plays a crucial role in cellular immunity, enhancing antigen processing and presentation promoting T-cell differentiation, facilitating leukocyte transport, and boosting antibacterial activity. Similarly, IL-4 and IL-10 are vital for driving T-cell and B-cell proliferation and differentiation. Epitopes that were competent to induce IFN-γ, IL-4, and IL-10 production were further examined for allergenicity, antigenicity, and toxicity analysis.

### 2.4. Prediction of Linear B-Lymphocyte (LBL) Epitopes

B-cells serve as vital components of the body’s immune system, responsible for producing antibodies that confer long-lasting immunity. ABCpred “http://crdd.osdd.net/raghava/abcpred/ (accessed on 16 June 2024)” was employed to forecast 16-mer LBL epitopes, using a threshold set at 0.51. Linear B-cell epitopes were then further validated to predict the antigenicity, allergenicity, and toxicity assessment of shortlisted LBL epitopes [25].

### 2.5. Population Coverage Analysis of MHC Alleles

Variations in the distribution and expression of human leukocyte antigen (HLA) alleles among different populations and ethnic groups worldwide may influence the effectiveness of epitope-based vaccines [26]. The IEDB population coverage webserver “http://tools.iedb.org/population/ (accessed on 20 June 2024)” was utilized to evaluate candidate vaccine coverage among the population. Subsequently, a geographic map illustrating the global distribution of HLA alleles was generated using the R-programming language “R-4.4.0 https://cran.r-project.org/bin/windows/base/ (accessed on 25 June 2024)”. This analysis aimed to assess the worldwide coverage of HLA alleles across different continents.

### 2.6. Molecular Docking Analysis of CTL and HTL Epitopes with MHC Alleles

Docking of epitopes with their associated MHC alleles was performed to assess their potential presentation on the cell surface for detection by T-cells. MHC molecules such as HLA-A*02:01 (PDB: 7RTD), HLA-A*29:02 (PDB: 7TLT), HLA-A*03:01 (PDB: 7L1C), HLA-B*15:01 (PDB: 8ELG), HLA-DRB1*13:02 (PDB: 1FV1), HLA-DRB1*15:01 (PDB: 8TBP), HLA-DRB1*04:01 (PDB: 5LAX), HLA-DRB1*12:01 (PDB: 3PDO), HLA-DRB1*11:01 (PDB: 6CPL), and HLA-DRB1*01:01 (PDB: 5V4N) were docked with high-ranking CTL and HTL epitopes. Any additional peptides, ions, water, and ligands were eliminated using ChimeraX v1.8 before docking. Docking was conducted using the PepDock “https://galaxy.seoklab.org/cgi-bin/submit.cgi?type=PEPDOCK (accessed on 6 July 2024)” available on GalaxyWEB (https://galaxy.seoklab.org/). Finally, the docking score between epitope and alleles was computed using PRODIGY “https://rascar.science.uu.nl/prodigy/ (accessed on 7 July 2024)”.

### 2.7. Designing of the Multi-Epitope Vaccine Construct against YEZV

To design the vaccine construct, specific LBL, CTL, and HTL epitopes originating from the YEZV’s structural proteins were strategically selected. An adjuvant, chosen for its known interaction with viral glycoproteins, was incorporated using a TLR4 agonist. An adjuvant 50S ribosomal protein L7/L12 was connected to the vaccine’s antigenic moiety through a bifunctional linker EAAAK (Glutamic acid (E)—Alanine (A)—Alanine (A)—Alanine (A)—Lysine (K)). Distinct linkers AAY (Alanine (A)—Alanine (A)—Tyrosine (K)), KK (Lysine (K)—Lysine (K)), and GPGPG (Glycine (G)—Proline (P)—Glycine (G)—Proline (P)—Glycine (G)) were utilized to bind the HTL, CTL, and B-cell epitopes, ensuring precise epitope presentation and enhancing the overall stability and efficacy. Spacer linkers (HEYGAEALERAG) were placed between epitopes to serve as cleavage sites for optimal lysosomal and proteasomal processing. To potentiate immune responses, the Pan HLA DR-binding (PADRE) amino acid sequence was inserted into the C-terminus of the vaccine model, leveraging its immunomodulatory properties. This strategic incorporation aims to boost antibody reactions and ensure comprehensive immune priming against the YEZV.

### 2.8. Evaluation of Physicochemical Properties of the Design Vaccine Construct

Assessment of the physiochemical properties of the designed vaccine construct was conducted using the Protparam tool “https://web.expasy.org/protparam/ (accessed on 10 July 2024)”. Additionally, VaxiJen v2.0 “http://scratch.proteomics.ics.uci.edu/ (accessed on 15 July 2024)” was utilized to predict antigenicity. AllerTOP v2.0 “https://www.ddg-pharmfac.net/AllerTOP/ (accessed on 16 July 2024)” was employed to forecast the potential allergenicity of the vaccine construct, while SOLpro “https://scratch.proteomics.ics.uci.edu/ (accessed on 16 July 2024)” was used to check the potential solubility of the vaccine construct using a 0.5 threshold which indicates potential solubility. Transmembrane helices were predicted with the DeepTMHMM v2.0 program “https://dtu.biolib.com/DeepTMHMM (accessed on 16 July 2024)”. Moreover, the potential signal peptides were analyzed using the SignalP 4.1 tool “https://services.healthtech.dtu.dk/services/SignalP-4.1/ (accessed on 16 July 2024)” during the last phase of the vaccine construct’s design.

### 2.9. Vaccine Structures Prediction and Its Validation

PDBsum “https://www.ebi.ac.uk/thornton-srv/software/PDBsum1/ (accessed on 17 July 2024)” was utilized to forecast the secondary structure of the designed vaccine construct, and the I-TASSER tool “https://zhanggroup.org/I-TASSER/ (accessed on 18 July 2024)” was employed to construct a three-dimensional structure from amino acid sequences by employing threading templates and evaluating the predicted models’ validity using the confidence score, which ranges from −5 to 2. Higher C-scores indicate greater confidence in the model’s accuracy. Following refinement, the three-dimensional structure of the vaccine model was uploaded to the GalaxyRefine tool. After refinement, the three-dimensional structure of the designed vaccine construct was validated by using the ProSA-web “https://prosa.services.came.sbg.ac.at/prosa.php (accessed on 19 July 2024), ERRAT, and PROCHECK web servers “https://saves.mbi.ucla.edu/ (accessed on 19 July 2024)”.

### 2.10. Disulfide Bonds Engineering

To facilitate the folding of the protein and to improve structural stability, the Disulfide by Design server “http://cptweb.cpt.wayne.edu/DbD2/ (accessed on 20 July 2024)” was employed for the engineering of a disulfide bond in the vaccine construct. For residue pairs meeting specific criteria, the Cα-Cβ-Sγ angle and χ^3^ value were set as default options. The value of chi^3^ ranges from −87° to +97° ± 30 and an energy value of <2.0 kcal/mol was selected [27]. Furthermore, the quality of the model containing specific disulfide bonds was evaluated using the ERRAT and ProSA-web.

### 2.11. Protein-Protein Docking

The human toll-like receptor-4 (TLR4) receptor was retrieved from the protein data bank (PDB) (PDB ID: 3FXI). All non-TLR chains were eliminated using ChimeraX v.1.8. The Dockprep tool in ChimeraX v.1.8 was employed with default parameters to prepare the TLR4 and vaccine construct for docking. The HADDOCK 2.2 “https://rascar.science.uu.nl/ (accessed on 20 July 2024) was utilized for docking studies. The HADDOCK refinement application was utilized to enhance the docking and entire positioning of the vaccine–TLR4 complex. Interface residues and intermolecular binding interactions among the chains within the complex of the vaccine and TLR4 were assessed by utilizing the PDBsum tool. The Gibbs free energy change for the complex was calculated by employing the PRODIGY web server with default values.

### 2.12. Evaluating Vaccine Construct Mobility and Stability through NMA Analysis

The modeled vaccine construct underwent a normal mode analysis (NMA) by using the iMODS webserver (https://imods.iqf.csic.es/) to visualize the internal angles and assess the overall functional movements of the vaccine. The dynamic simulation system in iMODS ensured minimal energy, stability of molecules, and activity of atoms in the designed vaccine construct within the docked complex [28].

### 2.13. Molecular Dynamic Simulation

The modeled vaccine, TLR4 receptor, and complex (TLR4+ vaccine) were simulated using Amber v20 software. The force field, such as ff14SB, was employed in the Amber 20 package. To achieve enough solvation and neutralization of every system, each system was solvated in a TIP3P water box, and sodium ions were added to balance the system charges. The energy was then reduced using the steepest descent minimization technique. Afterward, the temperature was elevated to 300 K. A Langevin thermostat was used to keep the temperature constant. Then, the equilibrated systems were subjected to the MD simulation using PMEMD. CUDA.

### 2.14. Immune Simulation

The C-ImmSim webserver “https://kraken.iac.rm.cnr.it/C-IMMSIM/index.php (accessed on 21 July 2024)” was utilized to verify the vaccine’s capability to stimulate the host immune system. The immune system’s response involves three anatomical components: the bone marrow, thymus, and lymph nodes. For the simulation of immunity, the parameters were set at volume 10, a random seed of 12345, 100 steps, 3 injections (spaced four weeks apart), and specific HLA types (A0101, B0702, A0101, B0702, DRB1_0101, DRB1_0101), with other settings left at their default values.

### 2.15. Predicting Vaccine mRNA Secondary Structure

Two online servers, Mfold v2.3 “http://www.unafold.org/mfold/applications/rna-folding-form-v2.php (accessed on 21 July 2024)” and RNAfold “http://rna.tbi.univie.ac.at/cgi-bin/RNAWebSuite/RNAfold.cgi (accessed on 21 July 2024)”, were employed to forecast the secondary structure of the vaccine mRNA. The primary outcome was the minimum free energy (ΔG Kcal/mol), focusing on lower values showing a more stable mRNA folding structure.

## 3. Results

### 3.1. Proteome Retrieval and Its Screening

The complete proteome of the Yezo virus (YEZV), comprising 60 proteins from global strains, was obtained from NCBI in the FASTA format (Appendix A), and CD-HIT was used to reduce redundancy, resulting in three unique proteins after performing BLASTp against the human proteome. The antigenicity, allergenicity, and toxicity assessment indicated high antigenicity and the absence of allergenic and toxic properties in all three proteins, which were chosen for further epitope prediction analyses (Figure 2 and Table 1).

### 3.2. Screening of Epitopes

Following the evaluation of antigenicity, allergenicity, toxicity, and the ability to stimulate IFN-γ, IL-4, and IL-10 secretion, three target structural proteins were selected to identify lead epitopes for the designing of a chimeric vaccine against the YEZV. From the predicted three proteins, the top rank eighteen epitopes (six LBL, six CTL, and six HTL) were chosen considering allergenicity, antigenicity, toxicity, and their capacity to induce IFN-γ (Table 2). The main objective was to find potential epitopes able to trigger both humoral and cell-mediated immune responses, while also triggering host interferons.

### 3.3. Global Coverage of Predicted Immunogenic Epitopes

The examination of the MHC-class I and II epitopes showed that the MHC-I epitopes cover 97.79% of the global population, whereas the MHC-II epitopes cover 67.76% of the global population. Given that a multi-epitope vaccine protein incorporates both MHC epitope classes, their collective assessment was used to ascertain the extent of population coverage. In total, 99.29% of the global population was encompassed. In Europe, the total coverage of MHC epitopes was 99.72%, followed by North America (99.66%), East Asia (99.11%), West Indies (98.37%), West Africa (97.93%), Oceania (97.36%), South Asia (97.76%), North Africa (98.47%), Southeast Asia (95.05%), Central Africa (95.48%), South Africa (95.64%), Northeast Asia (96.06%), East Africa (96.27%), Southwest Asia (95.24%), South America (95.04%), and Central America (47.03%). Evaluating the population coverage of the MHC (class I and II) epitopes, along with aggregate MHC epitopes, are depicted in Figure 3, Appendix A.

### 3.4. Molecular Docking Analysis of CTL and HTL Epitopes with HLA Alleles

Docking of the highest-scoring epitopes with their associating HLA alleles was analyzed to assess their ability to trigger the activity of helper T-cells. Docking revealed favorable molecular interactions between the CTL and HTL epitopes with their associating HLA alleles (Figure 4), and the interaction showed negative binding (ΔG ≤ −5.8 kcal/mol, average = −10.8 kcal/mol) (Appendix A), indicating stable interactions and anticipating a robust CTL and HTL immune response.

### 3.5. Formulation and Designing of Vaccine Construct

The vaccine construct was developed by integrating the top-raking LBL, CTL, and HTL epitopes, using KK, AAY, and GPGPG, (HEYGAEALERAG) spacer linkers, and the PADRE sequence. The adjuvant identified as 50S ribosomal protein L7/L12 was conjugated to the *N*-terminus of the vaccine construct using an EAAAK linker. During the construction of a vaccine, six CTL epitopes, six HTL epitopes, and six LBL epitopes were selected from the target structural proteins of the YEZV. Following meticulous combination and randomization processes, a vaccine consisting of 467 amino acid residues was constructed (Figure 5).

### 3.6. Physiochemical Properties of the Designed Vaccine Construct

To evaluate the vaccine constructs’ potential to bind to immune cell receptors, VaxiJen2.0 and ANTIGENpro projected antigenicity scores of 0.5747 and 0.6353, respectively. Both ToxinPred and AllerTOP servers ensure the non-toxic and non-allergic aspects of the vaccine construct. The physical and chemical analysis depicts the vaccine construct composed of 467 amino acids, with a molecular weight of 49.7 kDa, and a theoretical pI of 8.80. Its instability index of 24.22 indicated stability, while the aliphatic index of 92.72 suggested thermostability with a GRAVY value of 0.045. The calculated half-life in mammalian reticulocytes, yeasts, and *E. coli* was over 30, 20, and 10 h, respectively. The SOLpro server predicted a 96.44% probability that the vaccine construct would dissolve upon overexpression in *E. coli*. Furthermore, the lack of signal peptides and transmembrane helices suggests that there should be no complications in expression during the in vivo/in vitro analysis (Appendix A and Table 3).

### 3.7. 2D and 3D Structure Prediction, Refinement, and Its Validation

The 2D structure of the vaccine construct includes 30 helices, 58 beta turns, 33 gamma turns, and 39 helix–helix interactions (Figure 6A). I-TASSER generated five structural models for the design of the vaccine construct. The predicted confidence score values for templates −5 to 2 were −1.88, −1.91, −2.24, −2.18, and −2.37, respectively. Therefore, model 1 was identified by a confidence score of −1.88, exhibited a TM-score of 0.49 ± 0.15, a calculated RMSD of 11.6 ± 4.5Å, and was chosen for further analysis (Figure 6B). The GalaxyRefine tool resulted in five refined structures (Appendix A). Model 3 was identified as the best-refined structure, with Rama favored (88.0), poor rotamers (1.2), MolProbity score of (2.409), clash score (18.4), RMSD (0.449), and a GDT-HA score of (0.9400). The analysis of the Ramachandran plot indicates that 84.2% of the amino acids in the model were in favored regions, with 10.1% in additionally allowed regions, 0.8% in generously allowed regions, and 5.0% in disallowed regions (Figure 6C). The refined model had a Z-score of −3.04 (Figure 6D), and ERRAT estimated the quality factor to be 80.8% (Figure 6E).

### 3.8. Engineering of Disulfide Bond and Evaluation of Mutant Vaccine Construct

During disulfide engineering, 34 pairs of residues were pinpointed as potential candidates for disulfide mutation. After analyzing the χ^3^ angles and energy scores, five pairs of amino acids fulfilled the requirements with χ^3^ angles falling in the range of −87° to +97° and energy scores below 2.2 kcal/mol. Consequently, ten mutations were generated on the residue pairs LYS3-THR35, PHE13-PRO103, MET16-PHE24, GLY215-PHE281, and THR242-A246, with corresponding χ^3^ angles of 92.82°, 72.58°, 88.64°, 80.32°, and −74.68°, and energy scores of 2.06, 1.47, 1.72, 1.02, and 0.84 kcal/mol, respectively. The ERRAT score and ProSA−web analyses confirmed the high quality of the model with the introduced disulfide bonds (Appendix A).

### 3.9. Docking of Vaccine Construct with TLR4 Receptor

Docking revealed high affinity among the vaccine and TLR4 receptor, as indicated by low HADDOCK scores. Notably, cluster 3 exhibited a Z-score of −1.5, a HADDOCK score of −78.7 ± 9.9, a cluster size of 14, van der Waals energy of −34.0 ± 7.5, electrostatic energy of −312.8 ± 26.7, desolvation energy of 11.0 ± 4.1, restraints violation energy of 69.8 ± 11.6, and a significant buried surface area of 1520.8 ± 199.8. The root mean square deviation from the conformation with the lowest energy was 2.4 ± 2.1, and the Gibbs free energy (ΔG) was −9.9 kcal/mol. Consequently, the structure with the highest potential in cluster 3 was selected for molecular dynamics simulation. Docking analysis demonstrated the vaccine’s robust binding capability to the TLR4 protein. The association between the vaccine construct and TLR4 revealed the formation of eight hydrogen bonds and three salt bridges. Most residues in the vaccine construct that participated in salt bridge formation were LYS94-GLU278, ASP95-LYS224, and GLU92-LYS224, while some involved in hydrogen bond formation were LYS91-THR284, LYS94-GLU278, ASP98-GLN248, LYS102-THR245, LYS102-ILE218, ASP95-GLY221, ASP95-LYS224, and GLU92-LYS224 (Figure 7).

### 3.10. TLR4 Receptor and Vaccine Construct Interactions and Stability: NMA Insights

Normal mode analysis assessed the durability and molecular motions of the vaccine-TLR4 complex (Figure 8). The deformability analysis graph reveals prominent peaks, identifying regions where significant deformations occur within the primary chain residues of the molecular complex, and identifying potential hinges or linkers. (Figure 8A). The B-factor plot experimentally illustrated that NMA mobility correlates with the vaccine-TLR4 complex, reflecting average RMSD values of the docked structure (Figure 8B). The calculated eigenvalue of 1.128656e^−5^ suggested the rigidity linked to each natural mode of the vaccine-TLR4 complex (Figure 8C). Each normal mode was depicted individually (purple) and cumulatively (green) in the variance bar, illustrating an inverse relationship between variance and eigenvalue (Figure 8D). The covariance map visualized atomic motions within the complex, showing correlations (red), lack of correlations (white), and anti-correlations (blue) among different residue pairs (Figure 8E). An elastic network map depicted atom pairs connected by springs, highlighting their interaction and stiffness with darker shades indicating rigidity (Figure 8F). Overall, the NMA analysis suggested stable interactions between TLR4 and the chosen vaccine model.

### 3.11. Molecular Dynamics Analysis of Protein-Protein Interactions

#### RMSD Analysis

A total of 100 ns MD simulations were conducted for the TLR4 receptor and vaccine construct. The TLR4 receptor indicates small fluctuations during the first 60 ns, then the system moves toward stability. From 60 to 100 ns, the system revealed stable behavior. Minor deviations were seen from 105 to 120 ns, and then the system converged. During the last 10 ns, the system revealed high fluctuations. The average RMSD of the TLR4 receptor was found to be 1.4 nm. A similar pattern of RMSD was observed for the vaccine construct. Some minor deviations during 5–70 ns were observed. The system was found to be stable except for the last 10 ns. As compared to the TLR4 system and vaccine construct, the complex system was more stable during the entire 100 ns MD simulation. Initially, the RMSD of the complex was found to be 0.8 nm, then after 60 ns, the RMSD decreased to 0.7 ns and remained consistent till 100 ns. Overall the RMSD analysis confirmed that the vaccine model made a strong complex with the TLR4 receptor and was found to be highly stable throughout the 100 ns MD run (Figure 9).

### 3.12. RMSF Analysis

To evaluate the fluctuations at the residues level, an RMSF analysis was carried out. The TLR−4 in complex with vaccine model revealed less fluctuation as compared to the TLR4 and vaccine model in the Apo state. In the TLR4−vaccine complex system, residues such as 90–110, 300–310, and 401–405 revealed high fluctuations during the 100 ns MD run, while all other residues showed stable behavior, as shown in Figure 10. The RMSF analysis was consistent with the RMSD analysis which indicated the stability of the complex.

### 3.13. Virtual Immunogenicity Assessment

The designed vaccine model demonstrated a significant enhancement of secondary immune responses, as forecasted by computational immune stimulation. The initial simulated reaction displayed high IgM levels. The second and third simulations indicated a significant rise in B-cell numbers and high levels of IgG1, IgG2, IgM, and combined IgM + IgG antibodies, along with reduced antigen levels (Figure 11A,B). The increased number of memory B-cells and evidence of isotype switching indicated the development of immunological memory. Successive exposures to chimeric antigens resulted in a rapid decline in antigen levels (Figure 11C). It is hypothesized that further antigen exposures would result in the formation of memory in both (Tc) and (Th) cell subsets (Figure 11D,E). During the vaccination period, sustained high activity levels were observed in macrophages and natural killer cells (Figure 11F–H), as well as elevated cytokine levels including IL-2 and IFN-γ (Figure 11I). These findings support the hypothesis that the proposed vaccine can induce robust immune responses against the YEZV. Given these notable results, a plausible mechanism of action for the engineered vaccine was generated (Figure 12). The vaccine construct, designed from the structural virulent sequence, interacts with specific MHCs and TLR receptors, triggering the activation of key immune responses against this viral infection. Vaccination stimulated the proliferation of CTL, HTL, and other regulatory immune cells such as interferons, cytokines, and natural killer cells, aiming to eradicate the targeted viral infection.

### 3.14. Prediction of the Vaccine mRNA Secondary Structure

A vaccine mRNA two-dimensional structure was determined to have a minimum free energy of −408.40 kcal/mol (Figure 13B), with the centroid secondary structure exhibiting a minimum free energy of −329.28 kcal/mol (Figure 13C). The mountain plot provides a clear depiction of the thermodynamic ensemble for both the mRNA structure and its centroid structure (Figure 13A). The Mfold analysis determined the minimum free energy of the secondary structure to be −362.10 kcal/mol (Figure 13D). Since lower minimum free energy values correspond to greater mRNA stability, the vaccine mRNA is anticipated to be stable after in vivo expression.

## 4. Discussion

Addressing the growing global concern over the spread of the YEZV and the absence of effective vaccines, herein, an advanced immunoinformatics approach was employed by analyzing the virus structural proteins to design a vaccine construct to prevent future YEZV outbreaks. The identification of CTL, HTL, and LBL epitopes is crucial in generating a protective immune response that can effectively neutralize the virus and bolster long-term immunity [29,30,31]. Epitopes that overlap between CTL, HTL, and LBL, known for being highly antigenic, non-allergenic, and non-toxic, were prioritized due to their ability to trigger a humoral response. This response aided in fighting infections by either eliminating infected cells or producing substances, including antibodies and cytokines, involved in the immune response to establish long-term immunity [32,33,34]. To enhance this outcome, a vaccine construct was developed by combining various CTL, HTL, and LBL epitopes with specific linkers and adjuvants. The designed vaccine included EAAAK, AAY, HEYGAEALERAG, and GPGPG linkers and adjuvants to improve the vaccine’s structure and stability. Disulfide bonds between two cysteine residues help to maintain the protein’s three-dimensional shape, which is crucial for its stability and function [35]. These bonds were engineered to potentially enhance the vaccine’s proper shape, overall stability, and effectiveness in stimulating the immune system to provoke a strong immune response [36]. Molecular docking was performed to study the interaction between the vaccine constructs and human toll-like receptor-4 (TLR4) [37,38,39], which is important for activating immune cells and identifying viral peptide structures [40,41,42].

The obtained results indicated that the YEZV vaccine construct demonstrates high affinities for the TLR4 receptor, suggesting that the designed vaccine can potentially generate strong immune responses. Evaluating the vaccine’s ability to provoke an immune response involved simulating essential components of the mammalian immune system and tracking the response of various immune cells [43,44]. The designed vaccine construct potentially triggered vital immune components, such as antibodies (IgG and IgM), T-cells, B-cells, and cytokines. This multi-epitope-based subunit vaccine demonstrates protection against the YEZV. In a previous study, Asn282 and Glu278 residues of the TLR4 have been shown to interact with the vaccine construct, and molecular dynamic (MD) simulation identified Thr508 as the most fluctuating residue, while Gly384 remained the most stable residue throughout the simulation [45]. Similarly, another study demonstrated that the Asp95 residue of the vaccine construct was involved in interactions with the TLR4 receptor [46]. Our findings also highlight the involvement of Asn282, Glu278, and Asp95 residues in interactions, along with the significant fluctuation of Thr508 during the MD simulation. The molecular dynamics simulation and normal mode analysis suggested that the designed vaccine is stable. The immunological simulation indicated that the proposed vaccine construct has the capacity to trigger both cell−mediated and humoral immune responses within the host immune system. Future experimental confirmation is needed to establish the potency and dependability of the vaccine, including the synthesis of vaccine proteins and in vivo and in vitro testing.

## 5. Conclusions

The YEZV poses a significant public health challenge due to increasing susceptibility, underdiagnosis, and lack of awareness. Immunoinformatics approaches were utilized to design a vaccine targeting the YEZV’s structural proteins with enhanced antigenicity and verified non-allergenic and non-toxic properties. Molecular docking and MD simulations demonstrated that the vaccine effectively binds to various innate immune receptors and maintains stability under physiological conditions. Additionally, the vaccine elicited both cellular and humoral immune responses, indicating protective immunity against the YEZV. Thorough in vivo and in vitro testing in animal models is crucial to assess its efficacy and safety.

## Figures and Tables

**Figure 1 viruses-16-01408-f001:**
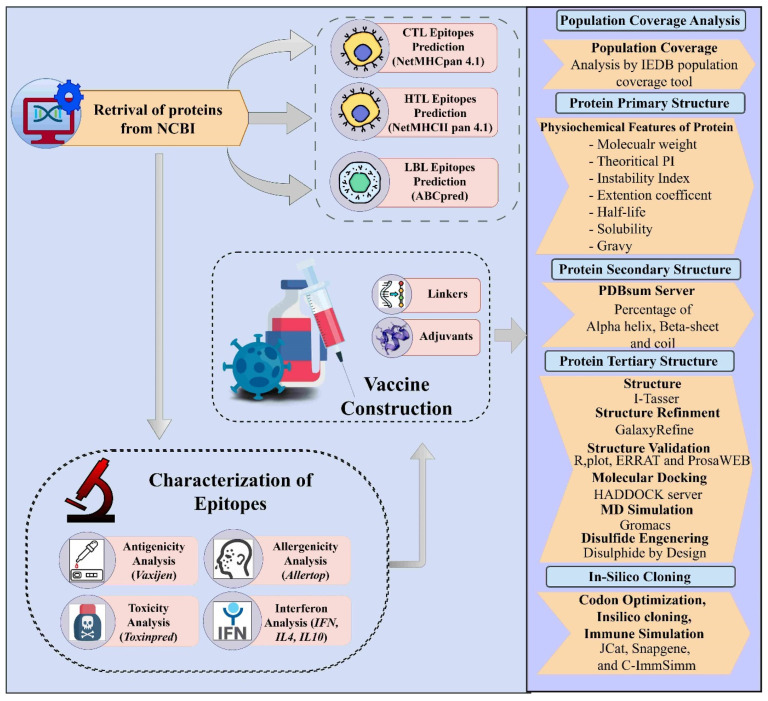
Chart depicting the computational immunoinformatics strategy for designing a multi-epitope vaccine construct against the Yezo virus.

**Figure 2 viruses-16-01408-f002:**
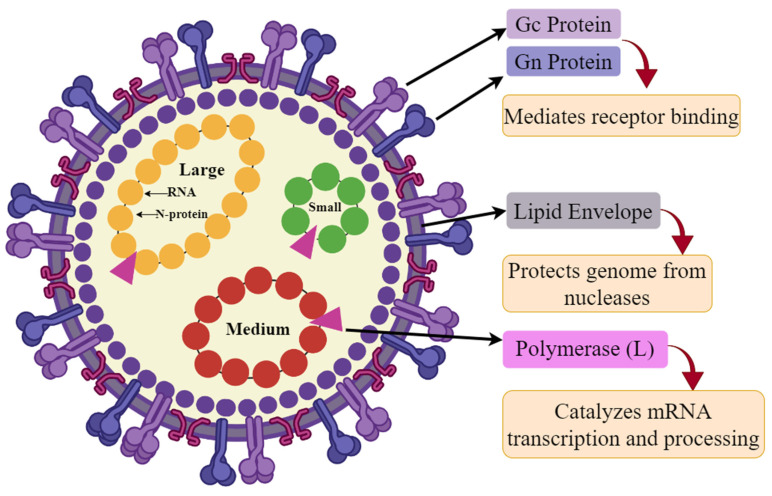
Genomic organization of the Yezo virus representing the structural proteins, created with Smart (https://smart.servier.com/).

**Figure 3 viruses-16-01408-f003:**
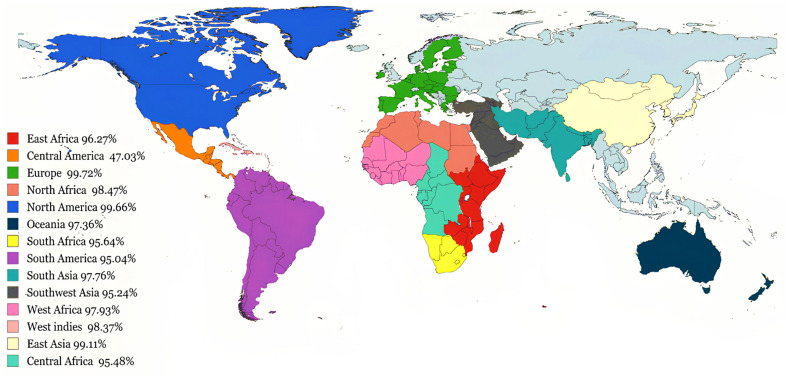
Population coverage across different countries and ethnicities.

**Figure 4 viruses-16-01408-f004:**
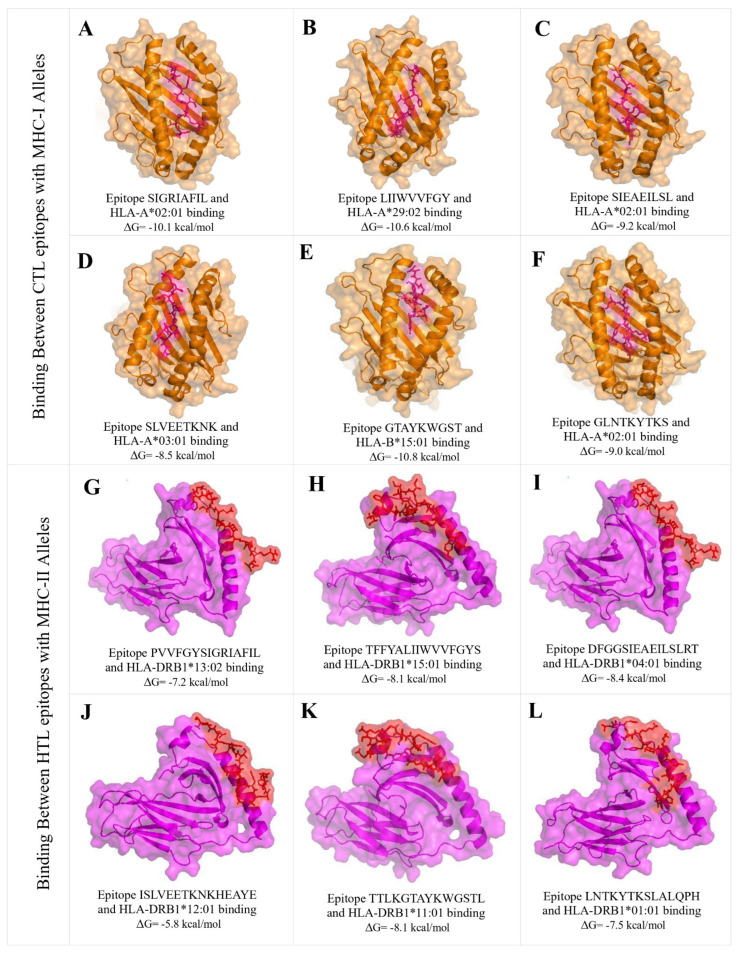
Docking was conducted for the top twelve MHC−I and II epitopes with their associating HLA allele binders. Panels (**A**–**F**) and (**G**–**L**) illustrate the docking results for the top six MHC−I epitopes with CTL epitopes and the top six MHC−II epitopes with HTL epitopes, respectively. The brown and purple structures in the figure symbolize the HLA alleles, while the magenta and red stick structures represent the corresponding epitopes.

**Figure 5 viruses-16-01408-f005:**
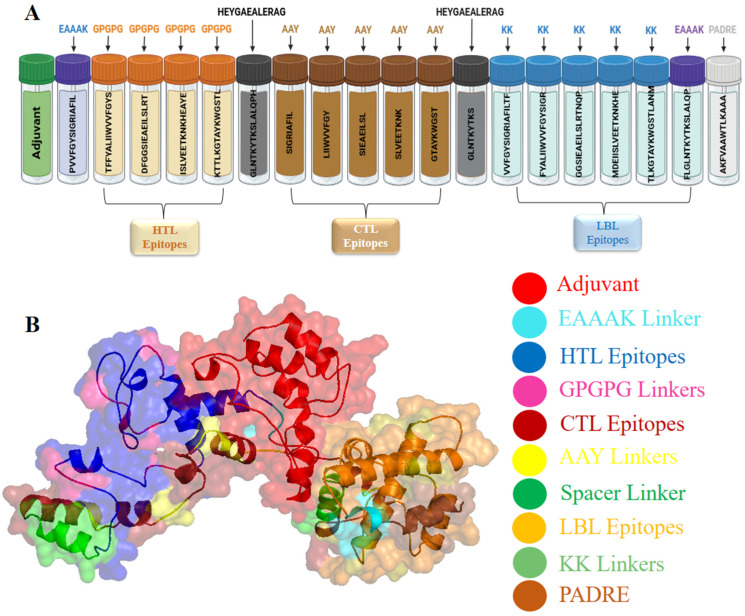
The figure (**A**) depicts the vaccine construct. (**B**) Illustrates the arrangement comprising an adjuvant and epitopes for CTL, HTL, and LBL, respectively. Notably, the adjuvant (red) and the primary CTL epitope (dark red) are connected via the EAAAK linker (light blue), while the HTL epitopes (dark blue) are conjoined using GPGPG linkers (purple), CTL epitopes are bridged by AAY linkers (yellow), LBL epitopes (light brown) are interconnected via KK linkers (light green), and PADRE is colored dark brown.

**Figure 6 viruses-16-01408-f006:**
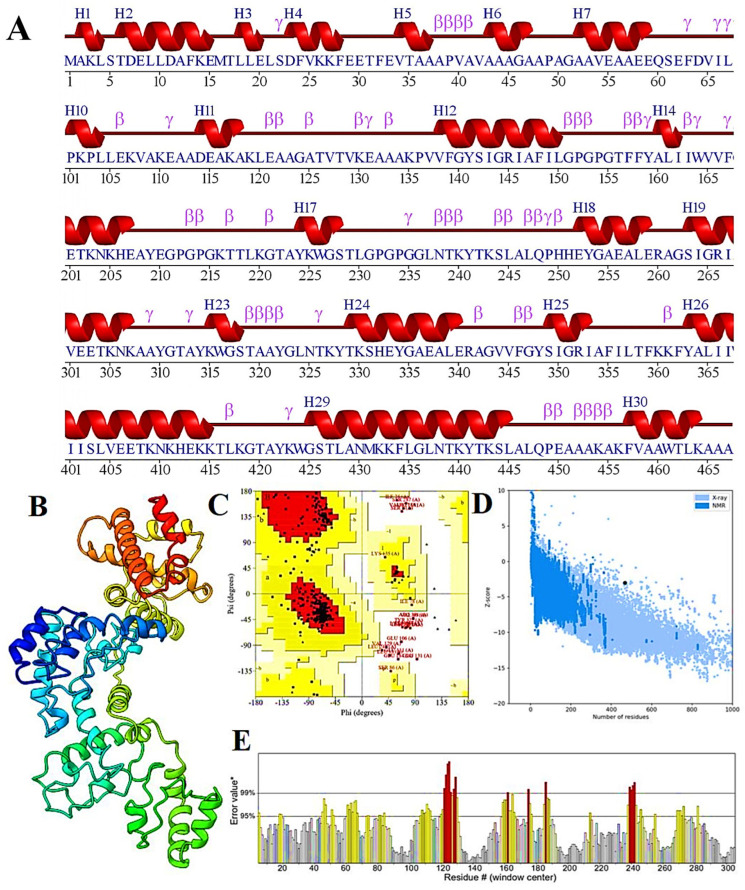
(**A**) Displays the predicted secondary structure generated using PDBsum. (**B**) Shows the rainbow color three-dimensional structure of the vaccine construct. (**C**) Presents a Ramachandran plot from the PROCHECK server, where red regions denote favored conformations, dark yellow and light yellow indicate additional allowed regions, and white represents disallowed regions. (**D**) Depicts a Z−score plot from the ProSA−web server, assessing the overall quality of the 3D model refinement. (**E**) Displays the ERRAT score, which evaluates the quality of the refined model. Regions with error rates exceeding 95% are highlighted in yellow, while residues with error rates surpassing 99% are exceptionally rare and are marked in red.

**Figure 7 viruses-16-01408-f007:**
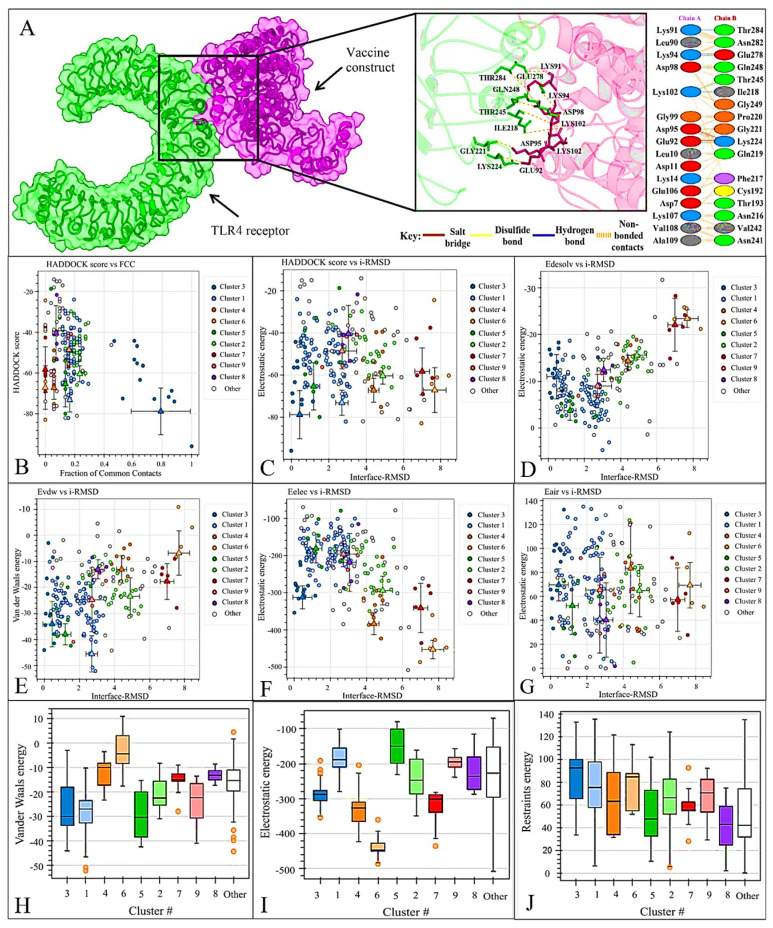
(**A**) Visualization of the vaccine and TLR4 complex and interacting residues. Chain A refers to the TLR4 receptor (green) and chain B refers to the YEZV vaccine construct depicted in purple. (**B**) Haddock scored against a fraction of frequent contacts. (**C**) Haddock score against ligand RMSD. (**D**) Electrostatic Solvation Energy (EDESOLV) against Initial−RMSD in Molecular Simulations (i−RMSD). (**E**) Van der Waals energy against interface i−RMSD. (**F**) Electrostatic energy (Eelec) of the docked molecule against interface RMSD. (**G**) (Ensemble−Averaged Interaction−Reweighted Simulation) EAIR outperforms i−RMSD. (**H**) Shows van der Waals energy plot. (**I**) Depicts electrostatic energy. (**J**) Shows restrained energy in predicting the structure of receptor-ligand complexes.

**Figure 8 viruses-16-01408-f008:**
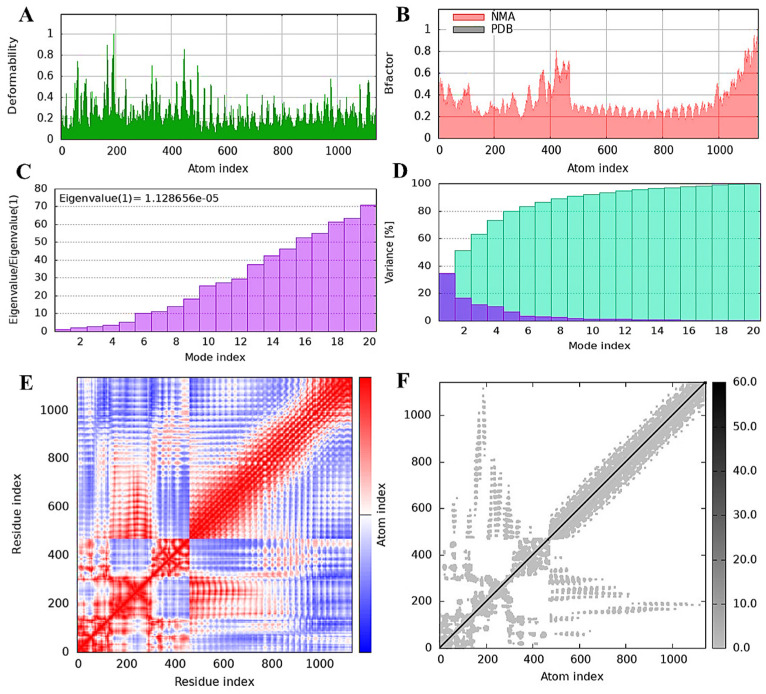
(**A**) Illustrates the flexibility of the main chain in the vaccine–TLR4 complex, with a focus on residue deformations that indicate the location of hinges and linkers. The B-factor plot in (**B**) shows the correlation of PDB fields with NMA mobility in the complex. The eigenvalue in (**C**) indicates the stiffness of motion in each normal mode. (**D**) Displays a variance map that illustrates the individual (blue) and cumulative (green) variances. The covariance graph in (**E**) highlights mobility relationships among residue pairs in the complex, showing correlated (red), uncorrelated (white), and anti-correlated (blue) movements. The elastic network model in (**F**) illustrates atom pairs connected by springs, with darker grey dots indicating greater spring stiffness.

**Figure 9 viruses-16-01408-f009:**
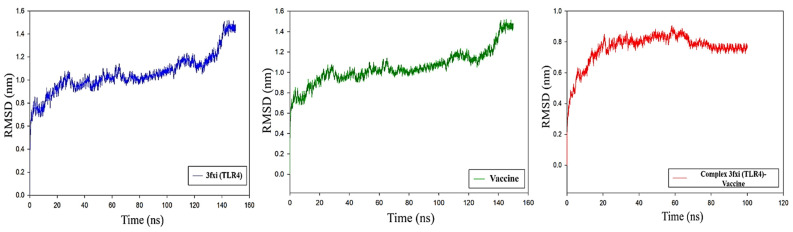
RMSD plots for TLR4 receptor (blue), vaccine (green), and TLR4−vaccine complex (red). The *x*-axis shows time in nanoseconds while the RMSD value in nm is shown on the *y*-axis.

**Figure 10 viruses-16-01408-f010:**
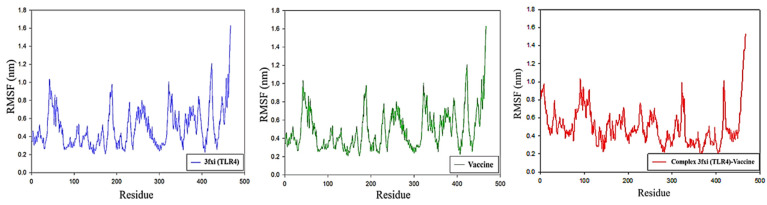
RMSF plots for TLR4 receptor (blue), vaccine (green), and TLR4−vaccine complex (red). The *x*-axis shows the residues number while the RMSF value in nm is shown on the *y*-axis.

**Figure 11 viruses-16-01408-f011:**
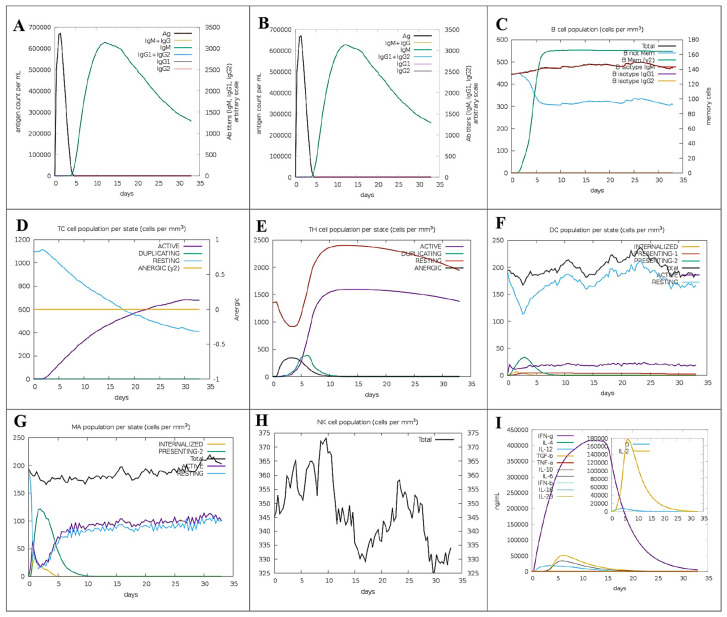
Computational immune simulation estimates the immunological potential of the YEZV recombinant peptide vaccine. Vaccination leads to an increase in levels of immunoglobulin antibodies and a decrease in levels of antigens. (**A**) Following immunization, B−cell numbers rise and antigen titers decrease (**B**). Repeated antigen exposure leads to further increases in B−cell counts (**C**). Tc and Th cell levels rise with each subsequent exposure to the antigen (**D**,**E**). Throughout the vaccination period, there is an elevation in macrophage, dendritic cell, and natural killer cell numbers (**F**–**H**). Furthermore, increased antigen exposure results in heightened production of cytokines and interleukins (**I**). This immune response, illustrated with leukocytes and the IL-2 expansion rate factor in the inset graphic, indicates a robust immunological activation.

**Figure 12 viruses-16-01408-f012:**
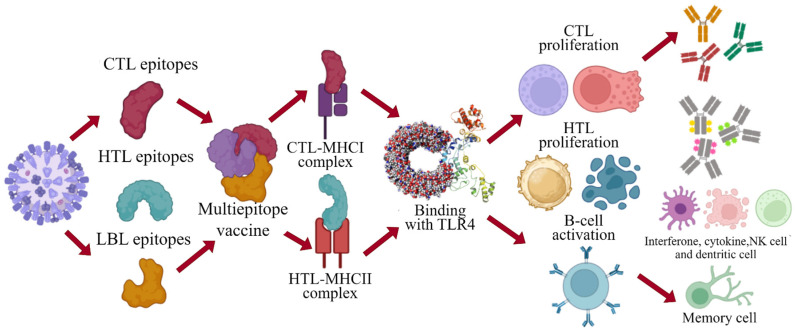
The proposed mode of action of the designed vaccine aims to stimulate a robust immune response by activating critical elements of the immune system, such as B−cells, T−cells, and regulatory cells. This activation is intended to effectively identify and eliminate viral pathogens, created with Smart.

**Figure 13 viruses-16-01408-f013:**
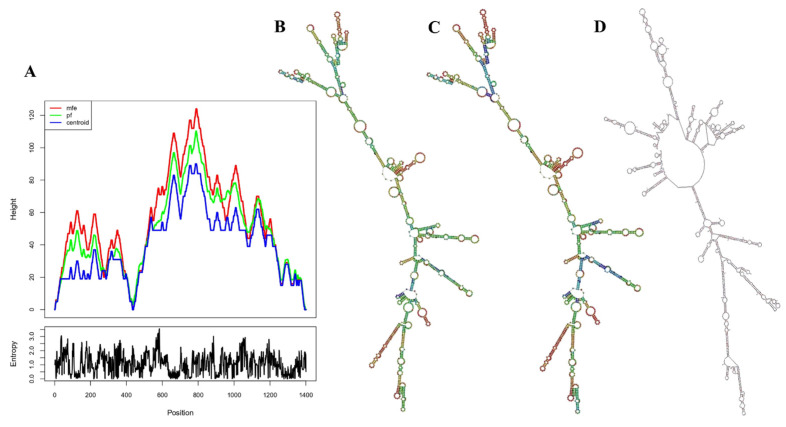
(**A**) Depiction of the thermodynamic state of the mRNA structure and its centroid form through a mountain plot. The positional entropy plot offers an in−depth understanding of every position. (**B**) The vaccine mRNA centroid 2D has a minimum free energy of −407.80 kcal/mol. (**C**) The optimal vaccine mRNA secondary structure with a minimum free energy of −474.30 kcal/mol. (**D**) The best 2D structure of the vaccine mRNA is identified through Mfold.

**Table 1 viruses-16-01408-t001:** The potential structural proteins for designing a multi-epitope vaccine construct targeting the Yezo virus.

Protein ID	Protein Name	Abbreviation	Antigenicity	Allergenicity	Toxicity
YP_010840879.1	Envelope glycoprotein	GP	0.4510	Non-allergic	Non-toxic
YP_010840880.1	RNA-directed RNA polymerase	L	0.4327	Non-allergic	Non-toxic
YP_010840881.1	Nucleoprotein	NP	0.4087	Non-allergic	Non-toxic

**Table 2 viruses-16-01408-t002:** Prioritized LBL, CTL, and HTL epitopes predicted by ABCpred and IEDB server.

Protein ID	LBL Epitopes	Score	Antigenicity/ Allergenicity/Toxicity	CTL Epitopes	Alleles	IC50	Antigenicity/ Allergenicity/Toxicity	HTL Epitopes	Alleles	Antigenicity/ Allergenicity/Toxicity	IC50	IFN-γ	IL-4	IL-10
YP_010840879.1	VVFGYSIGRIAFILTF	0.59	0.7885/NA/NT *	SIGRIAFIL	HLA-A*32:01, HLA-A*02:01, HLA-C*15:02, HLA-A*02:06, HLA-B*08:01	5	0.8896/NA/NT	PVVFGYSIGRIAFIL	HLA-DRB1*15:01, HLA-DRB1*13:02, HLA-DRB1*12:01	0.9977/NA/NT	59	+	+	+
	FYALIIWVVFGYSIGR	0.57	0.4648/NA/NT	LIIWVVFGY	HLA-A*29:02, HLA-A*26:01, HLA-B*15:02, HLA-A*30:02, HLA-A*25:01, HLA-B*35:01	1.4	0.8744/NA/NT	TFFYALIIWVVFGYS	HLA-DRB1*15:01	0.4551/NA/NT	88	+	+	+
YP_010840880.1	GGSIEAEILSLRTNQP	0.91	0.6088/NA/NT	SIEAEILSL	HLA-C*08:02, HLA-C*05:01, HLA-A*02:06, HLA-A*02:01	8.1	0.8887/NA/NT	DFGGSIEAEILSLRT	HLA-DRB1*12:01, HLA-DRB1*04:01	0.7948/NA/NT	71	+	+	+
	MDEIISLVEETKNKHE	0.85	0.7700/NA/NT	SLVEETKNK	HLA-A*03:01, HLA-A*11:01, HLA-A*68:01, HLA-A*30:01, HLA-A*31:01, HLA-A*30:02, HLA-B*46:01, HLA-B*15:01, HLA-A*02:01, HLA-A*26:01, HLA-A*32:01	7.9	0.8892/NA/NT	ISLVEETKNKHEAYE	HLA-DRB1*15:01, HLA-DRB1*12:01	1.1296/NA/NT	37	+	+	+
YP_010840881.1	TLKGTAYKWGSTLANM	0.91	0.4680/NA/NT	GTAYKWGST	HLA-A*30:01, HLA-A*30:02, HLA-B*15:01	10	0.5146/NA/NT	KTTLKGTAYKWGSTL	HLA-DRB1*08:02, HLA-DRB1*11:01, HLA-DRB1*15:01, HLA-DRB1*07:01	0.4351/NA/NT	70	+	+	+
	FLGLNTKYTKSLALQP	0.67	1.2806/NA/NT	GLNTKYTKS	HLA-A*02:01	9.1	1.3243/NA/NT	GLNTKYTKSLALQPH	HLA-DRB5*01:01, HLA-DRB1*04:01, HLA-DRB1*01:01	1.0808/NA/NT	26	+	+	+

* NA/NT: Non-Allergic, Non-Toxic.

**Table 3 viruses-16-01408-t003:** Physical and chemical features of the design vaccine construct.

S.No	Physicochemical Properties	Results
1	Total amino acids residue	467
2	Molecular weight	49,737.41
3	Extinction coefficients (at 280 nm in water)	69,790 M^−1^ cm^−1^
4	Theoretical pI	8.80
5	Formula	C_2290_H_3596_N_570_O_656_S_4_
6	Instability index	24.22 (Stable)
7	Aliphatic index	92.72
8	GRAVY value	0.045
9	Estimated half-life (mammalian reticulocytes, in vitro), (*E. coli* in vivo), (yeast in vivo)	>30 h, >10 h, >20 h
10	Antigenicity (AntigenPRO)	0.6255
11	Antigenicity (Vaxijen)	0.5904
12	Allergenicity (AllerTOP)	Non-Allergic
13	Solubility (Solpro)	0.964

## Data Availability

The datasets to support the conclusions of this article are given within the article.

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
