# Peer review of "Targeting Yezo Virus Structural Proteins for Multi-Epitope Vaccine Design Using Immunoinformatics Approach"

_viruses, 2024, doi:10.3390/v16091408_

Round 1

Reviewer 1 Report

Comments and Suggestions for Authors

The study by Rahman et al. uses informatics to identify immune hotspots in the structural proteins of YEZV (L, GP and NP). The ultimate aim was to identify new angles for vaccine development through assessment of key epitopes that can elicit both T- and B- cell responses. A multi-epitope vaccine was computationally designed based on these analyses from overlapping epitopes identified for CTLs, HTL, and LBLs. The authors predict that this should provide high protection against globally distributed strains which is highlighted very nicely in the world map schematic. This is an emerging and interesting area, and the paper is well written. Some of the wording in the figures isn’t quite as sharp as the text which could be addressed.

Overall, I think it holds value for acceptance with the caveat that without experimental confirmation, wording on computational data should be not so matter a fact, as these are predictions and still not experimentally proven data. Some comments are below to improve the manuscript are provided.

1. The threshold IC50s ≤100 nM used to select the epitope cut offs for the viral proteins investigated. Why this affinity? Can this be explained in more detail for a lay reader?

2. Figure 2: Wording is a little clunky describing the role of the viral proteins. This could be more succinct e.g. “mediates entry” “protection from nucleases” “catalyses not catalysed” – please revise.

3. In Table 1, presumably all proteins were assessed and the 3 were selected based on potential antigenicity. Can the full complement of viral proteins however be listed for comparison to justify the selection of the three. Also how do these compare to other viruses where this technique has been employed? Or were there other reasons for the specific selection?

4. The proteins are deemed not toxic, but polymerases are in this readers experience when overexpressed in cells. Can the authors make any comment to firm this up? What is meant be toxicity?

5. To the lay reader it is not clear as to how the ability to stimulate IFN-γ, IL-4, and IL-10 secretion were scored and ranked. Can these details be added? Is there evidence in the literature that these predictions are accurate when assessed experimentally?

6. Given this is limited to computational methods some of the statements do need to be dampened down. For example, it is stated that the codon optimisation shows “optimal expression in E. coli K12 480 vector”.  That is not actually experimentally shown and is a prediction. Statements such as “can generate an immune response” should state “can potentially generate” an immune response.

7. For some of the figures it would be good to show how these interactions compare relative to other viruses. For example, the MD simulations, can the TLR4 interactions be compared to a simulation from a viral peptide known to be recognised by TLR4 and run in parallel to show its comparable?

Minor comments

Lines 302-303 – I could not follow. Can this be explained in more detail?
Lines 305-306 – what were the randomisation processes?
Lines 511-512 vague – “anti-viral substances?”
Lines 518-519 – Can some virus examples be listed that are recognised by TLR4 to justify its selection. Also see the MD comment.
Table 3 should state h not hrs. What is a good half-life generally for this approach?
Lines 351-360 – can the rationale behind introduction of the disulphide bonds be explained to the lay reader.
Does showing the plasmid map in Figure 13B add any value?

Comments on the Quality of English Language

Only minor editing of English language required.

Author Response

Reviewer #1:

  1. The threshold IC50s ≤100 nM used to select the epitope cut offs for the viral proteins investigated. Why this affinity? Can this be explained in more detail for a lay reader?

Reply: Dear reviewer, many thanks for all your comments. We have explained the use of the IC50 ≤100 nM threshold lines 116-117 in the revised manuscript. This threshold was chosen to ensure that only epitopes with high binding affinity to MHC molecules are selected, as these are most likely to elicit a strong immune response.

  1. Figure 2: Wording is a little clunky describing the role of the viral proteins. This could be more succinct e.g. “mediates entry” “protection from nucleases” “catalyses not catalysed” – please revise.

Reply: We have revised the captions in Figure 2 for clarity as suggested.

  1. In Table 1, presumably all proteins were assessed and the 3 were selected based on potential antigenicity. Can the full complement of viral proteins however be listed for comparison to justify the selection of the three. Also how do these compare to other viruses where this technique has been employed? Or were there other reasons for the specific selection?

Reply: We assessed all 60 proteins identified from different strains of the Yezo virus reported from different countries. To eliminate paralogy and ensure non-redundancy, we applied CD-HIT clustering, which narrowed down the selection to three unique proteins. These proteins were subsequently filtered based on antigenicity, allergenicity, and toxicity analyses. The Envelope glycoprotein, RNA-directed RNA polymerase, and Nucleoprotein, listed in Table 1, were identified as the most promising candidates. To address the reviewer's request, we included a supplementary table S1 (line no:253) listing all the viral proteins for comparison to justify the selection process. This approach is in line with methodologies employed in similar immunoinformatics studies of other viruses, as demonstrated: DOI: 10.3389/fimmu.2023.1284366, DOI: https://doi.org/10.1038/s41598-024-53048-6.

  1. The proteins are deemed not toxic, but polymerases are in this readers experience when overexpressed in cells. Can the authors make any comment to firm this up? What is meant be toxicity?

Response: In our study, toxicity refers to the potential harmful effects of the protein under normal physiological conditions. RNA-dependent RNA polymerase (RdRp) proteins themselves are not inherently toxic under the normal conditions, as supported by Bressanelli et al. (2002), who discussed the role of RdRp in viral replication without evidence of inherent toxicity at physiological levels (DOI: 10.1128/JVI.76.7.3482-3492.2002). Our study primarily focused on predicting epitopes for vaccine development rather than exploring overexpression effects. We recognize the potential for toxicity under overexpression conditions and suggest that this consideration be taken into account in future experimental validations. To further substantiate the safety and viability of using RdRp in multi-epitope vaccine constructs, we have added references that successfully utilized RdRp protein for insilico vaccine development in similar contexts: DOI: 10.1186/s40203-015-0011-4, 10.3390/vaccines10101734, 10.3390/vaccines10101660.

  1. To the lay reader it is not clear as to how the ability to stimulate IFN-γ, IL-4, and IL-10 secretion were scored and ranked. Can these details be added? Is there evidence in the literature that these predictions are accurate when assessed experimentally?

Reply: The ability of epitopes to stimulate IFN-γ, IL-4, and IL-10 secretion was assessed using computational tools that predict cytokine responses based on epitope characteristics and MHC interactions. These predictions were scored and ranked accordingly. We have included details about the ability of epitopes to stimulate these cytokines in lines 129-132 of the revised manuscript. To support the accuracy of these predictions, we refer to the studies: DOI: 10.4103/abr.abr_10_17, https://doi.org/10.1371/journal.pone.0223844, https://doi.org/10.1007/s10529-021-03143-9, https://doi.org/10.1007/s12033-019-00227-w, which provide evidence that similar predictions validated experimentally. We plan to conduct further experimental validation in future work to confirm our findings.

  1. Given this is limited to computational methods some of the statements do need to be dampened down. For example, it is stated that the codon optimisation shows “optimal expression in E. coli K12 480 vector”. That is not actually experimentally shown and is a prediction. Statements such as “can generate an immune response” should state “can potentially generate” an immune response.

Reply: We have carefully revised the manuscript to better reflect the computational nature of our results / predictions. The statements regarding the optimal expression in E. coli, we have removed it as it does not add significant value to the overall presentation of the data. Additionally, we have revised the phrase "can generate an immune response" to "can potentially generate an immune response" to convey the predictive nature of our findings which is highlighted in line 512 in the revised manuscript.

  1. For some of the figures it would be good to show how these interactions compare relative to other viruses. For example, the MD simulations, can the TLR4 interactions be compared to a simulation from a viral peptide known to be recognised by TLR4 and run in parallel to show its comparable?

Reply: The docking and MD simulations studies are now compared with the previous studies and highlighted in line 518-525 in the revised manuscript.

Minor comments

Lines 302-303I could not follow. Can this be explained in more detail?

Response: Now corrected.

Lines 305-306 – what were the randomisation processes?

Response: Randomization process involved systematically selecting and arranging six CTL, six HTL, and six LBL epitopes from the target structural proteins to ensure diverse representation. The combinations were randomized to create a vaccine construct with a balanced and optimized sequence.

Lines 511-512 vague – “anti-viral substances?”

Response: “anti-viral substances” refers to antibodies and other immune molecules that neutralize or inhibit the virus. The mistake in now corrected which is highlighted in line 497 in the revised manuscript.

Lines 518-519Can some virus examples be listed that are recognised by TLR4 to justify its selection. Also see the MD comment.

Response: We added references of those studies that performed immunoinformatics analyses and molecular docking of multi-epitope vaccine constructs with TLR4, also MD to support our approach, such as monkeypox virus, Respiratory syncytial virus, and Tick-borne encephalitis virus, in lines 508 of the revised manuscript. We also mention those research paper which validate the insilico analysis on viruses like; (Bovine leukemia virus https://doi.org/10.1371/journal.pone.0199397, and Hepatitis B virus https://doi.org/10.1128/jvi.79.11.7269-7272.2005) experimentally by in vitro/in vivo analysis.

Table 3 should state h not hrs. What is a good half-life generally for this approach?

Response: We have corrected Table 3 to state “h” instead of “hrs” in the revised manuscript. Regarding the question about a "good" half-life, it is generally considered that a longer half-life indicates greater stability and persistence of the protein (Volker et al., 2009; https://doi.org/10.1038/nbt.1588). For therapeutic proteins or vaccine candidates, a half-life greater than 30 hours is often desirable to ensure effective functionality and stability in vivo. In our study, the estimated half-lives are >30 hours in mammalian reticulocytes, >10 hours in E. coli, and >20 hours in yeast, which suggest good stability across different expression systems. These values indicate that the proteins are relatively stable, which is beneficial for their potential application as vaccine candidates.

Lines 351-360 can the rationale behind introduction of the disulphide bonds be explained to the lay reader.

Response: According to the kind reviewer’s comment, we added explanation of disulphide bonds in line 502-506 in the revised manuscript.

Does showing the plasmid map in Figure 13B add any value?

Response: Thank you for your suggestion. Upon reconsideration, we have removed Figure 13B.

Reviewer 2 Report

Comments and Suggestions for Authors

This is quite a novel paper in that it presents no "wet science" to support the theoretical conclusions.  Thus, it lacks a punchline to either confirm or deny the suggestion of an effective vaccine design from in silico analysis of virus protein sequences.  

Despite this weakness, I do think that the paper offers an intriguing approach to vaccine design that deserves serious consideration.  

However, some of the conclusions regarding the choice of epitopes from different virus proteins is  surprising.  Epitopes from a virus envelope glycoprotein I can understand.  Even ones from a nucleoprotein are understandable in light of reports about similar virus proteins have potential as vaccine candidates.  The RNA polymerase is a little more difficult to understand since I don't know of any other similar proteins that have been suggested as vaccines for viruses diseases. It could be that we have all been guilty of preconceived notions as to what a vaccine should comprise.  Almost all virus vaccines are based on a surface protein, although some may comprise whole virus particles, attenuated or inactivated.

I return to my comments above about lack of experimental validation.  The authors might have least have made a plasmid construct and tested expression in E. coli. Either way, this would have tested the approach taken and informed on future work, even if the expression levels were not great.  The speed with which genes can be assembled these days could enable the work to be done within a month or so.  I hope that this avenue is soon to be explored.

Comments on the Quality of English Language

The quality of English is good, with only minor edits required by journal.

Author Response

Reviewer #2:

This is quite a novel paper in that it presents no "wet science" to support the theoretical conclusions. Thus, it lacks a punchline to either confirm or deny the suggestion of an effective vaccine design from in silico analysis of virus protein sequences. Despite this weakness, I do think that the paper offers an intriguing approach to vaccine design that deserves serious consideration. However, some of the conclusions regarding the choice of epitopes from different virus proteins is surprising. Epitopes from a virus envelope glycoprotein I can understand. Even ones from a nucleoprotein are understandable in light of reports about similar virus proteins have potential as vaccine candidates. The RNA polymerase is a little more difficult to understand since I don't know of any other similar proteins that have been suggested as vaccines for viruses’ diseases. It could be that we have all been guilty of preconceived notions as to what a vaccine should comprise. Almost all virus vaccines are based on a surface protein, although some may comprise whole virus particles, attenuated or inactivated. I return to my comments above about lack of experimental validation. The authors might have least have made a plasmid construct and tested expression in E. coli. Either way, this would have tested the approach taken and informed on future work, even if the expression levels were not great. The speed with which genes can be assembled these days could enable the work to be done within a month or so. I hope that this avenue is soon to be explored.

Response: Dear Reviewer, thank you for your comment. We acknowledge the importance of experimental validation to support our theoretical conclusions. While our current study focuses on in silico analysis due to limitations in experimental resources and budget constraints, we recognize the need for practical verification. We plan to address these aspects, including plasmid construction and expression in E. coli, in future research. Regarding the choice of epitopes, including those from RNA polymerase, we understand your concerns. Recent studies have explored non-surface proteins, like RdRp, for their potential in vaccine development. We have cited relevant literature supporting the inclusion of RNA polymerase in epitope-based vaccines (DOI: 10.1186/s40203-015-0011-4, 10.3390/vaccines10101734, 10.3390/vaccines10101660).

Round 2

Reviewer 1 Report

Comments and Suggestions for Authors

N/A

Comments on the Quality of English Language

N/A